# Patterns of Food Assistance Program Participation, Food Insecurity, and Pantry Use among U.S. Households with Children during the COVID-19 Pandemic

**DOI:** 10.3390/nu14050988

**Published:** 2022-02-26

**Authors:** Kaitlyn Harper, Emily H. Belarmino, Francesco Acciai, Farryl Bertmann, Punam Ohri-Vachaspati

**Affiliations:** 1Department of International Health, Johns Hopkins Bloomberg School of Public Health, Johns Hopkins University, 615 N Wolfe Street, Baltimore, MD 21205, USA; 2Department of Nutrition and Food Sciences, University of Vermont, 109 Carrigan Drive, Burlington, VT 05405, USA; emily.belarmino@uvm.edu (E.H.B.); fbertman@uvm.edu (F.B.); 3Food Systems Program, University of Vermont, 109 Carrigan Drive, Burlington, VT 05405, USA; 4College of Health Solutions, Arizona State University, 500 N 3rd Street, Phoenix, AZ 85004, USA; facciai@asu.edu (F.A.); punam.ohri-vachaspati@asu.edu (P.O.-V.)

**Keywords:** federal nutrition assistance programs, COVID-19, child food security

## Abstract

This study aims to describe differences in participation in the Supplemental Nutrition Assistance Program (SNAP), Special Supplemental Nutrition Program for Women and Children (WIC), and school meal programs by household characteristics prior to and during the pandemic, and to examine the association of program participation with food security status and food pantry use. We analyze secondary data (*n* = 470) from an online survey collected in July/August 2020 using weighted multiple logistic regression models. Participation in SNAP declined among households with children in the first four months of the pandemic, while participation in WIC increased slightly, and participation in school meals remained unchanged. There were significant differences in SNAP, WIC, and school meal programs use by race/ethnicity, income, and urbanicity before and during the pandemic. Food insecurity prevalence was higher among SNAP participants at both periods but the gap between participants and non-participants was smaller during the pandemic. Pantry use and food insecurity rates were consistently higher among federal nutrition assistance program participants, possibly suggesting unmet food needs. These results highlight the need for increased program benefits and improved access to food, particularly during periods of hardship.

## 1. Introduction

In 2019, prior to the start of the COVID-19 global pandemic, 13.6% of United States (U.S.) households with children (compared to 10.5% of all U.S. households) experienced food insecurity [1]. The COVID-19 pandemic disproportionately impacted these households as they faced challenges related to the closure of schools and childcare centers, in addition to widespread job losses and reductions to job-related income [2,3]. As a result, the prevalence of food insecurity in households with children was higher during the pandemic compared to all households [4]. Additionally, food insecurity in both households with adults and children has been found to be higher in communities of color due to long-standing structural and systemic inequities [4,5].

Food insecurity is detrimental to the physical and psycho-social health of both children and caregivers [6,7,8,9]. Children experiencing food insecurity are more likely to have poor diet quality [10], leading to increased risk of chronic diet-related disease. They also face adverse developmental, social, and academic outcomes, with potential long-term consequences [8,11]. In the U.S., federal food and nutrition assistance programs aim to reduce food insecurity by providing food or benefits for purchasing food. The largest of these programs is the Supplemental Nutrition Assistance Program (SNAP), which offers a monthly cash benefit for purchasing foods with the goal of alleviating food insecurity and supporting access to a healthy diet. SNAP is a means-tested program, and households are eligible to receive SNAP benefits if their gross monthly income is at or below 130% of the federal poverty level [12]. Additionally, households may qualify through categorical eligibility if they already receive benefits from other means-tested assistance programs [13]. Benefit amounts are calculated based on household income and number of people in the family. During the pandemic, these calculation rules were suspended due to emergency allotments, which raised most households’ SNAP benefits to the maximum amount [14] However, during the period under consideration, the benefits did not increase for households with the lowest incomes who were already at the maximum amount. These households did not receive additional benefits until the passage of subsequent relief packages [15]. Additionally, during this time, income from federal unemployment insurance was counted towards income, which resulted in some participants losing eligibility for SNAP [16]. In response to the pandemic, the U.S. Department of Agriculture (USDA) Food and Nutrition Services (FNS) worked with states and retailers to expand the online SNAP purchasing pilot, which was available in 47 states by the end of 2020 [17]. This rapid expansion aimed to provide greater access to food to SNAP participants; however, online redemptions only accounted for 3% of benefits redeemed in December 2020 [17].

The Special Supplemental Nutrition Program for Women, Infants, and Children (WIC) is another means-tested program that provides monthly supplemental nutritious food packages to pregnant, lactating, and post-partum individuals, as well as infants and children under 5 years [18]. WIC eligibility is based on gross income of no more than 185 percent of the federal poverty level or through categorical eligibility. During the pandemic, the USDA granted programmatic waivers that allowed WIC agencies flexibility in implementation for remote certification and enhancement of benefits [19].

School-aged children in low-income households may also receive free or reduced-price nutritious meals through the National School Lunch Program and School Breakfast Program (hereafter school meals) [20,21]. Children in households with incomes below 130 percent of the poverty level or those receiving other federal assistance programs qualify for free meals. Those with family incomes between 130 and 185 percent of the poverty line qualify for reduced-price meals [22]. Additionally, schools and school districts may apply for the Community Eligibility Provision (CEP) if 40% of enrolled children qualify for free- or reduced-price meals through categorical eligibility, which allows all children to receive school meals at no cost. In the 2019–2020 school year, over 30,000 schools from all 50 states and the District of Columbia participated in CEP [23]. During the pandemic, public schools were allowed to serve free grab-and-go meals to all children, regardless of eligibility [24].

Participation in federal nutrition assistance programs has been shown to reduce food insecurity in low-income families [25]. However, numerous barriers prevent some individuals from accessing these programs, including lack of information about the programs, difficulty applying, perception of stigma, and criteria restrictions [26,27,28]. While previous research has shown that participation in federal food and nutrition assistance programs increases during periods of economic hardship [29], pandemic-related barriers may have hampered program participation by some groups [30,31]. The use of food banks and pantries also increases in response to unmet food needs, as was evidenced by the long lines outside these facilities nationwide during the pandemic [32]. However, barriers, such as difficulty locating pantries and transportation, or perceived barriers around immigration status, hinder individuals’ ability to access pantries, particularly those in certain subpopulations [33,34,35].

This paper seeks to better understand the use of federal food and nutrition assistance programs among households with children during the COVID-19 pandemic. Specifically, we aim to (i) describe differences in participation in SNAP, WIC, and school meals programs by household characteristics prior to and during the first four months of the pandemic, and (ii) examine the associations of program participation with household food security status and food pantry use, during both time periods. We hypothesize that there are socio-demographic differences in program participation and that, after adjusting for income and other key covariates, respondents participating in federal nutrition assistance programs are less likely to experience food insecurity and less likely to use food pantries compared to non-program participants.

## 2. Materials and Methods

### 2.1. Sample Population

We surveyed 1510 U.S. adults in July/August 2020 using Qualtrics (Provo, UT, U.S.A.) online panels. Survey participants were selected from two online panels conducted in parallel: the first panel was representative of the U.S. adult population based on the 2018 American Community Survey data (5-year estimates) for race/ethnicity (hereafter race) and age, while the second was representative for race but lower-income households (i.e., with an annual household income in 2019 less than USD 50,000) were oversampled. The response rates of 10.4% and 27.0% were calculated for the national sample and low-income sample, respectively, using the American Association for Public Opinion Research’s calculator for web-based surveys (Response Rate 4) [36,37]. For the current analysis, which focuses on respondents in households with children, survey weights were calculated using income data for households with children from the 2019 national Census [38], so that results may be generalized to reflect the national income distribution of households with children.

### 2.2. Variables and Measures

The survey asked participants to answer questions on a variety of topics, including food security, food access, and food assistance program participation, as well as household and individual demographic characteristics. The period from March 2019 to 10 March 2020 was referred to as “prior to the pandemic”, while the period from 11 March 2020 to the time of the survey (covering approximately four months) was referred to as “since the onset of the pandemic”. The average time for completing the survey was 23 min. Fifty-seven percent of respondents identified as female, and respondent age was distributed equally across three age groups: 18–34, 35–54, and 55+. Respondents in the analytical sample, described below, represented 47 states in the U.S. plus the District of Columbia.

The key variables in our analysis reflected use of federal nutrition and food assistance programs, specifically SNAP, WIC, and school meals. Respondents were asked to indicate which programs (i.e., SNAP, WIC, and school meals), if any, they used the year prior to and since the onset of the pandemic. Each federal program variable in the analysis indicates whether a respondent used the program, either independently or in conjunction with another program. For example, those counted in the SNAP category may have either only used SNAP or may have used SNAP in addition to WIC and/or school meals. The other two key variables in our analysis were food pantry use, as an indicator of unmet food needs, and food security status. Food security was measured using the USDA six-item module [39], in which respondents who say “yes” to two or more of the six questions are categorized as experiencing food insecurity. Food pantry use was measured by asking respondents if they used food pantries/food banks in the 12 months prior to the pandemic and since the onset of the pandemic.

Demographic variables collected in the survey and explored in the analysis included (1) household annual income in 2019 with six response categories that were regrouped into three for analysis: less than USD 50,000, USD 50,000–99,000, and greater than USD 100,000; (2) race/ethnicity, dichotomized as non-Hispanic white and non-white due to sample size limitations; (3) number of children in the household, dichotomized as one child versus more than one child; (4) urbanicity, captured using participant zip code of residence and categorized as rural and urban based on the classification system from the Rural Health Research Center [40]; and (5) U.S. Census region determined based on respondent state of residence [41]. The parent survey collected more granular data on race/ethnicity, but we were unable to disaggregate data in this secondary analysis due to sample size limitations. Additionally, in the current sample (of households with children), less than 4% of households reported four or more children. Almost all households reported one or two children, with ~10% of households reporting three children. Because of this considerable skewness in the distribution, the variable for number of children was dichotomized. To double check that this would not affect the results, a sensitivity analysis was run for the models presented, using number of children as a continuous variable. This did not substantially change any of the estimated coefficients or their standard errors, (all changes were approximately ±1%). Large rural, small rural, and isolated categories were combined into one rural category due to sample size limitations.

### 2.3. Analysis

Analyses were limited to 470 households with children under the age of 18 that had complete data for all relevant variables. Analyses were conducted in STATA 15, two-tailed p-values were used for all analyses, and statistical tests were considered significant at p-values less than 0.05. First, weighted distributions of variables were examined. Next, weighted logit multivariate models were used along with the “margins” command to calculate adjusted prevalence of SNAP, WIC, and school meals participation (overall and by household demographic characteristics) for the two time periods examined. These models included race, income category, number of children under 18 years of age, U.S. region, and urbanicity.

Lastly, a second set of multivariate logit regression analyses were used to test the association of program participation with two outcome variables—food insecurity and pantry use. These models included all three programs—SNAP, WIC, and school meals— as three separate binary variables, were adjusted for all covariates listed above, and were run for the two time periods analyzed, the year prior to the pandemic and the first four months of the pandemic. Based on these models, adjusted prevalence and 95% confidence intervals were calculated for each outcome both prior to and during the first four months of the pandemic.

## 3. Results

### 3.1. Description of Program Participants

Table 1 presents a description of the sample. Of the 470 respondents, the majority were white (73.2%), living in an urban area (90.2%), and with a household income over USD 100,000 (61.2%). About a quarter of respondents had a household income of USD 50,000–99,000 (27.2%) and 11.6% had a household income of less than USD 50,000. More than one-third of respondents were from the Southern Census regions (39.0%), about a quarter were from the Northeast (26.0%), and 14.4% and 20.6% were from the Midwest and West, respectively. In the 12 months prior to the pandemic, 37.4% of the households with children participated in SNAP compared to only 27.8% during the first four months of the pandemic. For WIC, 19.4% of households with children participated in the year prior to the pandemic compared to 24.3% during the first four months of the pandemic. For school meals, 24.7% participated prior to the pandemic compared to 22.8% during the first four months. A total of 22% and 28.8% of participants received food from food pantries prior to and during the first four months of the pandemic, respectively. Food insecurity rates increased from 38.7% to 51.8% over the two time periods under investigation.

### 3.2. Differences in Program Participation by Household Characteristics

Based on the results from the first set of multivariate models, the adjusted program participation rates—overall and by key household demographic characteristics—are presented in Figure 1. Overall, respondents were more likely to use SNAP the year prior to the pandemic than during the first four months of the pandemic. Conversely, respondents were less likely to use WIC prior to the pandemic compared to the first four months. There were no significant differences in overall school meal participation between the two time periods.

White participants were more likely to use SNAP prior to the pandemic compared to non-white participants and were also more likely to use WIC during both time periods. Those in urban areas were also more likely to use WIC during the pandemic than those in rural areas. Participants in the lowest income categories (<USD 50,000) were more likely to use SNAP during both time periods and WIC during the pandemic compared to those in the middle (USD 50,000–99,000) and highest (>USD 100,000) income categories. 

### 3.3. Association of Food Insecurity and Pantry Use with Program Participation

Based on the second set of multivariate analyses, 36.8% and 53.0% of households with children experienced food insecurity in the 12 months prior to and during the first four months of the pandemic, respectively. The adjusted prevalence of food insecurity by program participation is presented in Table 2. Both prior to and since the onset of the pandemic, households using SNAP were significantly more likely to experience food insecurity compared to households not using SNAP. There was no difference in the prevalence of food insecurity by WIC and school meals participation status prior to the pandemic. However, during the first four months of the pandemic, households using WIC and school meals were significantly more likely to experience food insecurity compared to those not using the programs.

The adjusted prevalence of food pantry use by program participation is presented in Table 3. Overall, 16.7% and 24.9% of respondents used food pantries prior to and during the first four months of the pandemic. Prior to the pandemic, households using SNAP, WIC, and school meals were more likely to use food pantries compared to households not using the programs. Similarly, during the first four months of the pandemic, households using SNAP and WIC were significantly more likely to use food pantries compared to those not using the programs.

## 4. Discussion

In this sample of households with children, we (1) examined the use of federal food and nutrition assistance programs in the year prior to and during the first four months of the COVID-19 pandemic by household characteristics, and (2) assessed the association of program participation with food insecurity and food pantry use. As hypothesized, we found significant differences in the use of SNAP, WIC, and school meal programs by race, income, and urbanicity before the pandemic; most of these differences persisted or were exacerbated in the months following the pandemic. Overall, compared to the year before the pandemic, SNAP participation declined among households with children in the first four months of the pandemic, while WIC participation increased slightly, and school meal participation remained unchanged.

Similar declines in SNAP participation among vulnerable households in the months following the onset of the pandemic have been observed by others [30,42]. During the pandemic, initial SNAP policies that counted additional sources of benefits (e.g., unemployment insurance) towards household income may have made some households with children ineligible. Ettinger de Cuba and colleagues (2019) found that SNAP benefit reductions and cutoff during economic hardship were a barrier to SNAP participation among households with children and that such reduction was associated with greater odds of poor health outcomes [43]. Further, while USDA SNAP waivers designed to facilitate enrollment were implemented early in the pandemic, technology demands and lack of transportation may have contributed to lower participation rates among some households with children [44]. A Utah-based study found that SNAP-eligible respondents experienced travel-related barriers to SNAP certification and recertification during a similar time period in 2020 [45].

National data reflect inconsistent shifts in WIC participation [46]; 26 states and the District of Columbia reported an increase or no change, and 24 states reported a decline in participation. In the current study, rural and urban households reported participating in WIC at similar rates in the year prior to the pandemic, but in the month following the start of the pandemic, significantly fewer rural households participated in WIC. Rural areas in the U.S. experience higher rates of poverty and have been shown to experience more challenges to WIC participation in pre-COVID-19 times [47]; these challenges may have heightened during the pandemic, resulting in lower rates of participation.

Prior to the pandemic, SNAP and WIC participation rates were higher among white households in this study. In the first few months of the pandemic, the rates became somewhat more equitable, but for different reasons. Fewer white households participated in SNAP in the first four months of the pandemic, while more non-white households participated in WIC. These disparate trends by geography and race call for investigating strategies and waivers that may be necessary for reaching different populations in periods of emergency.

Contrary to our hypothesis, in regression models adjusting for income and other covariates, food insecurity and food pantry use were significantly higher for SNAP users compared to non-SNAP users, both prior to and during the first four months of the pandemic. Examining the coefficients and confidence intervals for the two models for SNAP participation at these two points, we also note that, although the prevalence of food insecurity increased among all households, the increase was less pronounced among SNAP households (4.5 percentage points) compared to that for non-SNAP households (23.6 percentage points). We saw a similar pattern in food pantry use, where pantry use increased by 3.5 percentage points among SNAP users compared to 10.7 percentage points among non-SNAP users. However, since separate models were estimated for SNAP (and food pantry) use for periods prior to and during the pandemic, statistical comparisons of coefficients across models was not feasible. These findings suggest that households receiving SNAP benefits, though already experiencing higher rates of food insecurity and pantry use, may have been less likely to see a further exacerbation of their unmet needs during the pandemic, compared to non-SNAP households for whom we observed more pronounced changes in both food insecurity and pantry use during the first four months of the pandemic. These are important findings as they highlight the protective effect of SNAP participation during emergencies, and therefore the need to ensure the adequate reach of the program during such difficult times. WIC and school meals also showed differential patterns of food insecurity and pantry use among participants and non-participants. The considerable increase in food pantry use observed among households that did not participate in SNAP could be the result of unmet food needs experienced by these households). However, it could also indicate that, during the first four months of the pandemic, a greater number of disadvantaged households experienced barriers in accessing SNAP and thus had to resort to food pantries. This interpretation is in line with the results from a recent study, which found that SNAP participation among the most vulnerable—defined as low-income, food insecure—households declined during the first months of the pandemic [48].

A different pattern was observed for WIC, where both food insecurity and pantry use almost doubled among participating households, with much smaller increases observed among non-participating households. While WIC participation has consistently shown benefits to participants and their families in terms of improved diet quality [49], the lower amount of overall benefits and the challenges to obtain WIC-approved items during the pandemic may have contributed to higher unmet needs among participating households.

### 4.1. Limitations

This study is not without limitations. The data collected were cross-sectional and therefore we cannot assess the temporality of the association between food insecurity and program participation. For instance, the higher food insecurity rates among SNAP participants both prior to and during the first four months of the pandemic likely derive from the fact that households experiencing food insecurity are more likely to become SNAP recipients. The extent of this self-selection, which also applies to the other programs, is unknown in the current data. However, it is likely that it occurred similarly across the two time periods. Therefore, our comparisons of food insecurity rates and pantry use prevalence across the two time periods based on program participation status remain valid. Although recent examinations of online panels have been shown to be credible sources for collecting data [50], data collected from online panels may underrepresent those with low literacy, inability to take surveys in English/Spanish, or those without cell phones or access to the Internet. The data collected from the parent survey were representative of the U.S. population by race and income. However, because this secondary analysis used only a subset of data from the parent survey, our analysis was not nationally representative, and the results of this study may not be generalizable to the entire U.S. population of households with children. The parent survey also did not collect data on citizenship or whether respondents were born outside the U.S.

Additionally, the prevalence of food insecurity found in this study is substantially higher than the prevalence documented by the USDA [51]. One possible explanation of this may be timing of data collection; we collected data in the first four months of the pandemic, while the USDA collected data in December 2020, after many people received stimulus checks and other federal benefits, or after some people may have returned to work. Another limitation of this study is that program participation was self-reported and open to social desirability and recall bias. Recall bias is of particular concern given that the survey asked about two periods at one point in time. Social desirability bias would likely occur for both time points recalled, and therefore, would not affect the comparisons in participation over time. It is also important to note that, although program participation prior to the pandemic referred to a 12-month period, program participation since the onset of the pandemic only covered a 4-month period. This may have contributed to underestimate the rates of participation during the first four months of the pandemic. Finally, the USDA six-item food insecurity module does not explicitly ask food security questions about children in the household but was chosen to lower respondent burden; research shows that household food insecurity prevalence estimates differ only minimally from those derived from the 18- or the 10-item modules [39].

### 4.2. Public Health Implications

Our findings show that participation in three of the largest federal nutrition assistance programs was associated with significantly higher prevalence of food insecurity and food pantry use, both in the year prior and in the months following the pandemic. One potential explanation is that, while these programs have important benefits, they are not adequate to meet household needs, especially during emergencies. Prior research has shown that households in greater need and with limited access to food tend to utilize safety net programs more frequently, and that food insecurity persists among many participating households [1,6]. Thus, while the USDA’s means-tested food assistance programs strive to help eligible households meet their needs, participants may still rely on the charitable food system to fulfill the unmet needs during emergencies, in times of hardship or, in many instances, chronically [52,53]. Unlike SNAP, WIC, and school meal programs, which have income eligibility requirements, food pantries are usually accessible to all households. Food banks and pantries are critical to providing immediate solutions to severe food deprivation; thus, an increase in food pantry use is a useful indicator of unmet food needs in the community.

## Figures and Tables

**Figure 1 nutrients-14-00988-f001:**
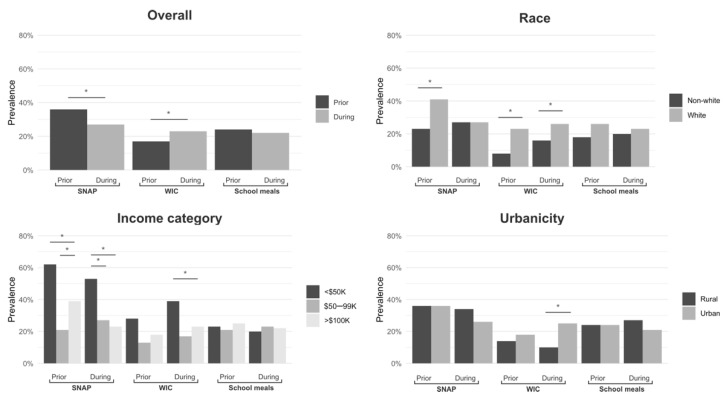
Adjusted prevalence for SNAP, WIC, and school meals participation in the 12 months prior to the pandemic by household demographic characteristics. Asterisks represent statistically significant differences. Abbreviations: Supplemental Nutrition Assistance Program (SNAP); Special Supplemental Nutrition Program for Women, Infants, and Children (WIC).

**Table 1 nutrients-14-00988-t001:** Demographic characteristics of households with children prior to and during the first four months of the pandemic (*n* = 470).

Sample Characteristics	% (*n* = 470)
Race/ethnicity ^a^	
Hispanic	13.8
Non-Hispanic White	73.2
Non-Hispanic Black	8.6
Asian	3.1
Native American	0.3
Multiple Races	1
Income	
<USD 50,000	11.6
USD 50,000–99,000	27.2
>USD 100,000	61.2
% households with multiple children (<18 years of age)	58.5
Urbanicity (% urban)	90.2
U.S. Census Region	
Northeast	26
Midwest	14.4
South	39
West	20.6
Food pantry use	
Prior to the pandemic	22.7
First four months of the pandemic	28.3
Food insecurity	
Prior to the pandemic	38.7
First four months of the pandemic	51.8

^a^ Race/ethnicity is dichotomized into “non-Hispanic White” and “non-White” in statistical models because of small sample sizes of all subgroups other than non-Hispanic White.

**Table 2 nutrients-14-00988-t002:** Adjusted prevalence of food insecurity among households with children by federal food assistance program participation status prior to and during the first four months of the pandemic.

	% Food Insecurity					
	Prior			During		
	%	95% CI		%	95% CI	
Overall	36.8	31.9, 41.6	-	53.0	47.9, 58.1	-
SNAP						
No	23.7	18.3, 29.1	<0.01	47.3	41.6, 53.1	<0.01
Yes	62.5	54.4, 70.5		67.0	57.4, 76.6	
WIC						
No	36.1	30.8, 41.4	0.63	45.8	40.3, 51.2	<0.01
Yes	39.7	26.5, 52.8		73.5	64.1, 83.0	
School Meals						
No	34.5	29.2, 39.8	0.14	48.9	43.3, 54.5	0.01
Yes	44.1	32.2, 56.0		66.2	54.9, 77.5	

Weighted logistic regression was used to evaluate significance. Covariates included race, income, number of children in household, region, and urbanicity. Abbreviations: Supplemental Nutrition Assistance Program (SNAP); Special Supplemental Nutrition Program for Women, Infants, and Children (WIC); Confidence interval (CI).

**Table 3 nutrients-14-00988-t003:** Adjusted prevalence of food pantry use among households with children by federal food assistance program participation status prior to and during the first four months of the pandemic.

	% Food Pantry Use					
	Prior			During		
	%	95% CI		%	95% CI	
Overall	16.7	12.7, 20.7		24.9	20.6, 29.2	
SNAP						
No	10.4	6.6, 14.2	<0.01	21.1	16.3, 25.8	<0.01
Yes	33.4	23.5, 43.4		36.9	26.8, 47.0	
WIC						
No	14.4	10.4, 18.4	<0.01	18.3	14.2, 22.4	<0.01
Yes	29.5	18.2, 40.8		53.2	42.2, 64.1	
School Meals						
No	13.1	9.2, 16.9	<0.01	22.6	18.0, 27.2	0.05
Yes	32.7	21.9, 43.5		34.0	22.8, 45.2	

Weighted logistic regression was used to evaluate significance. Covariates included race, income, number of children in household, region, and urbanicity. Abbreviations: Supplemental Nutrition Assistance Program (SNAP); Special Supplemental Nutrition Program for Women, Infants, and Children (WIC); Confidence interval (CI).

## Data Availability

The survey used in this study is publicly available and can be accessed at: https://dataverse.harvard.edu/dataverse/foodaccessandcoronavirus (Accessed on 1 June 2021).

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
