# Peer review of "Patterns of Food Assistance Program Participation, Food Insecurity, and Pantry Use among U.S. Households with Children during the COVID-19 Pandemic"

_nutrients, 2022, doi:10.3390/nu14050988_

Round 1

Reviewer 1 Report

The manuscript entitled, "Patterns of food assistance, program participation, food insecurity and pantry use among US households with children during the COVID-19 pandemic”, focuses on a critical topic – namely, the increased demand for nutrition assistance and higher levels of food insecurity among households with children in the U.S. during the pandemic. However, there are several major issues with the manuscript (i.e., methodological flaws, inadequate understanding of the multiple nutrition assistance programs mentioned), which are detailed further below:

Introduction

  1. The introduction does not adequately describe the nutrition assistance programs referenced. SNAP, WIC, and the free/reduced school meal programs are very different government nutrition assistance programs in terms of their target population/s and eligibility. The introduction does not even mention that SNAP is a means-tested program aimed at alleviating food insecurity among eligible low-income households. Meanwhile, the free/reduced school meal program has been available to entire schools or districts if they qualified for the Community Eligibility Provision as a result of changes from the Healthy, Hunger Free Kids Act of 2010, which means that higher income children may also be eligible if they attend a high poverty school. These programs also drastically changed during the pandemic (i.e., SNAP online purchasing expansion) and these changes are not well described. Further, there are different barriers to participation for different sub-populations.
  2. Information about food banks/pantries is also lacking in the introduction and what is known about participation and access to food pantries by different subpopulations (and their availability).
  3. Research has shown that racial/ethnic minorities in the U.S. were disproportionately impacted by increases in household food insecurity levels during the pandemic. This is not addressed or mentioned although it is an important facet. Further, racial/ethnic minority groups vary in their enrollment and participation of these programs, which is not mentioned.

Methods

  1. The binary white vs. non-white treatment of race is not appropriate given the heterogeneity of racial/ethnic groups and their differences in participation in nutrition assistance programs. The authors do not appropriately explain why they use a binary variable for race, nor is it mentioned in the limitations. It is a considerable limitation given the heterogeneity of races/ethnicities in the U.S. and regions where the sample was drawn.
  2. External validity issues - the sample limitations are considerable (particularly the availability of the survey only in English and the online panel recruitment given the large digital divide between racial/ethnic groups), so I am unsure how to assess the external validity and the results’ generalizability. How does the sample compare to the U.S. population of households with children? The authors characterize the sample as “nationally representative sample of households with children”, but this claim is questionable.
  3. Survey data collection – Respondents were asked about two time periods, but it seems as though they were only asked about these two periods at one point in time. Recall bias is of real concern. This bias is mentioned in the limitations, but it is an important issue that can impact the accuracy of the data collected. Is this approach supported or validated elsewhere? I have previously only reviewed data about two time periods collected at two time periods.
  4. The survey nonresponse rate is not provided.
  5. Grouping together nutrition assistance programs for this variable is inappropriate and is not justified in the manuscript. These are very different programs so participating in SNAP vs. SNAP+school meals vs. SNAP+WIC vs. SNAP+WIC+school meals are all very different circumstances. It is unclear why the authors grouped them together they did, which poses an issue for interpretation of the results.
  6. What theory is guiding the modeling? What are the hypotheses that are being tested?
  7. Why is number of children dichotomized versus being treated as a continuous variable?

Results/discussion

  1. The novel contributions of the results are unclear. For instance, the finding that participants in the lowest income categories are more likely to use SNAP both prior to and during the pandemic is associated with program eligibility—those at higher income levels do not qualify and it is unclear how income may have changed in the first four months of the pandemic (this data was not collected).
  2. The result focused on food pantry use is the most useful and novel, however the results on the associations between program participation and food pantry use are less novel and questionable due to the methodological issues identified.
  3. The claim that these analyses “shed light on how safety net program participation varied by race” is called into question because race/ethnicity are dichotomized into white and non-white.

Author Response

Response to Reviewer 1 Comments

Introduction

  1. The introduction does not adequately describe the nutrition assistance programs referenced. SNAP, WIC, and the free/reduced school meal programs are very different government nutrition assistance programs in terms of their target population/s and eligibility. The introduction does not even mention that SNAP is a means-tested program aimed at alleviating food insecurity among eligible low-income households. Meanwhile, the free/reduced school meal program has been available to entire schools or districts if they qualified for the Community Eligibility Provision as a result of changes from the Healthy, Hunger Free Kids Act of 2010, which means that higher income children may also be eligible if they attend a high poverty school. These programs also drastically changed during the pandemic (i.e., SNAP online purchasing expansion) and these changes are not well described. Further, there are different barriers to participation for different sub-populations.

Thank you for your helpful comment. We have added additional details to our introduction relevant to each of the three federal programs discussed in the paper. Please see lines 50-99.

  1. Information about food banks/pantries is also lacking in the introduction and what is known about participation and access to food pantries by different subpopulations (and their availability). Research has shown that racial/ethnic minorities in the U.S. were disproportionately impacted by increases in household food insecurity levels during the pandemic. This is not addressed or mentioned although it is an important facet. Further, racial/ethnic minority groups vary in their enrollment and participation of these programs, which is not mentioned.

We have added additional information about barriers to pantry use for subpopulations on lines 97-99 and additional information and references about increases in food insecurity among communities of color on lines 38-40.

Methods

  1. The binary white vs. non-white treatment of race is not appropriate given the heterogeneity of racial/ethnic groups and their differences in participation in nutrition assistance programs. The authors do not appropriately explain why they use a binary variable for race, nor is it mentioned in the limitations. It is a considerable limitation given the heterogeneity of races/ethnicities in the U.S. and regions where the sample was drawn.

We agree that dichotomization of race into white vs. non-white treatment is not ideal and should be actively avoided through equitable sampling. However, as noted in our methods section (lines 225-227), we were unable to completely disaggregate data on race/ethnicity due to sample size limitations in this secondary analysis. We have added a note on lines 415-417 of the discussion to acknowledge this limitation.

  1. External validity issues - the sample limitations are considerable (particularly the availability of the survey only in English and the online panel recruitment given the large digital divide between racial/ethnic groups), so I am unsure how to assess the external validity and the results’ generalizability. How does the sample compare to the U.S. population of households with children? The authors characterize the sample as “nationally representative sample of households with children”, but this claim is questionable.

We agree that the sample limitations do not allow us to draw conclusions that are generalizable to all US households with children. The parent survey collected is nationally representative by race and income; however, the secondary analysis is only representative based on household income (through inverse probability sampling weights). We have added a statement to lines 419-411 in the limitations to acknolwedge this. We have also removed “nationally representative” from the abstract and first line of the discussion.

  1. Survey data collection – Respondents were asked about two time periods, but it seems as though they were only asked about these two periods at one point in time. Recall bias is of real concern. This bias is mentioned in the limitations, but it is an important issue that can impact the accuracy of the data collected. Is this approach supported or validated elsewhere? I have previously only reviewed data about two time periods collected at two time periods.

We agree that recall bias can impact the accuracy of data and have added more detail to our description of biases in our limitations (lines 449-450). There are numerous others papers that have relied on responses for current and prior periods collected at one time point, including one recently published in Nutrients (doi: 10.3390/nu13030712) and another in Frontiers in Nutrition (doi: 10.3389/fnut.2021.647365). While there is potential for recall bias, we do not anticipate that it affects various groups examined differentially.

  1. The survey nonresponse rate is not provided.

We have added response rate for the parent survey on lines 187-189.

  1. Grouping together nutrition assistance programs for this variable is inappropriate and is not justified in the manuscript. These are very different programs so participating in SNAP vs. SNAP+school meals vs. SNAP+WIC vs. SNAP+WIC+school meals are all very different circumstances. It is unclear why the authors grouped them together they did, which poses an issue for interpretation of the results.

Thank you for your question. We agree that there are different circumstances for participating in, for example, SNAP only versus SNAP and other programs. We would like to clarify that we did not group all program participation indicators into one single variable. Instead, we included three separate predictors in our models: SNAP, WIC, and school meals participation—all binary variables coded 0=No; 1=Yes. This allowed for our estimates for SNAP to adjust for participation in WIC and school meals, and vice versa. One thing we did not do was examine joint participation using an interaction term, which would allow us to estimate an adjusted prevalence in food pantry use (or food security) for all combinations of program use. This approach, unfortunately, would require a larger sample size. In fact, using a three-way interaction term, there would be eight separate groups, with cell sizes of n=7, 20, 26, 32, 38, 47, 69, and 231. However, even if our models do not show potential interactive effects, we can still interpret the SNAP coefficient as the association between SNAP participation and food pantry use (or food security status), net of everything else, including participation in the other programs. To clarify this to the reader, we have added a short description to our methods section on lines 250-254.

  1. What theory is guiding the modeling? What are the hypotheses that are being tested?

Our hypotheses are stated at the bottom of the introduction section. We state, “We hypothesize that there are socio-demographic differences in program participation and that after adjusting for income and other key covariates, respondents participating in federal food and nutrition assistance programs are less likely to experience food insecurity and less likely to use food pantries compared to non-program participants.” Because the current paper is descriptive in scope, our modeling was guided by previous research rather than a theory. But if the reviewer has any specific suggestions, we will certainly consider them.

  1. Why is number of children dichotomized versus being treated as a continuous variable?

Thank you for your question. In our sample, <4% of households reported four or more children. Almost all households reported one or two children, with ~10% of households reporting three children. Because of this considerable skewness in the distribution, it was appropriate to dichotomize the number of children variable into “1” and “2 or more”. However, to double check that this would not affect our results, we ran a sensitivity analysis for each of the models presented using number of children as a continuous variable. This did not substantially change any of the estimated coefficients or their standard errors, (all changes were approximately ±1%), so we decided to keep the number of children as a dichotomous variable. We have noted this in our methods section on lines 229-238. 

Results/discussion

  1. The novel contributions of the results are unclear. For instance, the finding that participants in the lowest income categories are more likely to use SNAP both prior to and during the pandemic is associated with program eligibility—those at higher income levels do not qualify and it is unclear how income may have changed in the first four months of the pandemic (this data was not collected).

Thank you for your comment. Our study contributes to the growing body of literature addressing the effects of the COVID-19 pandemic on households with children. Specifically, we examine food assistance program participation among households with children during the first four months of the pandemic compared to the period prior to the onset of the pandemic. Our findings that SNAP participation decreased significantly among these at-risk households during the first four months is important. Further, we examine the impact of specific program participation on household food security status and unmet food needs as measured by food pantry use. While food insecurity rates increased among all households, the increase was less pronounced among SNAP households compared to non-SNAP households, adjusting for household income. We saw a similar effect on food pantry use, also suggesting that fewer SNAP households had unmet needs, compared to non-SNAP households at similar levels of income. These are important finding as they highlight the protective effect of SNAP participation during emergencies, and therefore the need to ensure adequate reach of the program during such difficult times. We have added additional details to our discussion highlighting the importance of this study on lines 348-355 and 408-416.

  1. The result focused on food pantry use is the most useful and novel, however the results on the associations between program participation and food pantry use are less novel and questionable due to the methodological issues identified.

We agree that the results focused on the food pantry are the most useful and novel. In the revised discussion, we tried to highlight the novelty of these results. If the methodological issues mentioned in the comment refer to the speficiation of some of the predictors (i.e., program participation, rac/ethnicity, number of children), we covered that in our responses to earlier comments and made the necessary changes in the text.

  1. The claim that these analyses “shed light on how safety net program participation varied by race” is called into question because race/ethnicity are dichotomized into white and non-white.

We agree with this comment and have removed the quoted statement from our discussion.

Reviewer 2 Report

This article strives to address changes in food assistance program and food pantry use  before (by retrospective recall)  and in the early days of the COVID epidemic among   an online standing panel of 470 Qualtric households with children.  This is a topic that others have sought to address. It is competently written, but has a number of methodologic challenges, some of which are well acknowledged by the authors in section 4.1 However, there are a number of other concerns:

The authors seem to be unaware of confounding by indication – it is an oft repeated finding that SNAP participants have higher rates of food insecurity (FI) than non-respondents because it is often FI that drives recipients to jump through the bureaucratic hoops to participate. SNAP benefits were  inadequate to purchase even the Thrifty Food Plan in most states  for a month’s worth of food before the 15% COVID American Recovery Act increase in Jan 2021 in benefits for those other than poorest who were already receiving maximum allotment for family size, and remains so even after the $.40 per meal per person increase in October 2021, which has been outstripped by inflation in food prices. 

increase in SNAP benefits for millions of families included in the Families First Coronavirus Response Act passed in March 2020. The initial structure of the policy change increasing benefits to the maximum amount meant those with the lowest incomes already at the maximum benefit amount did not see any increase until passage of subsequent relief packages, while pre-existing eligibility requirements meant that some otherwise eligible lawfully present immigrants who arrived in the U.S. within the past five years remained disqualified for the program.    ( https://www.cbpp.org/research/food-assistance/usda-states-must-act-swiftly-to-deliver-food-assistance-allowed-by:

https://www.fns.usda.gov/snap/eligibility/citizen/non-citizen-policy

 The inclusion of households with parents born out side the US is not described in the description of the Qualtrics panel. Since one in four children in the US have at least one immigrant parent this is a concerning omission.

2) The authors claim that their panel (which is 73% NonHispanic White people) is reflective of the US population as a whole, or of SNAP recipient families particularly in terms of race/ethnicity, is difficult to defend. 2020 Census data show that only 57.8% of US population as a whole identifies as non-Hispanic White.  By recent standards (JAMA. doi:10.1001/jama.2021.17579 based on US Preventive Service Task Force standards)  “non-White” is not an acceptable classification – racially minoritized is preferred, but this group should be further disaggregated in description of sample (Black, Hispanic or Latinx, Asian Native American.) even if subgroups are too small for analysis.  In terms of SNAP participation race/ethnicity also  differs considerably from the Qualtric sample.

New USDA Report Provides Picture of Who Participates in SNAP

https://frac.org › blog › new-usda-report-provides-picture

3) By defining children as those less that 18 years without further disaggregation by child age the authors have an inaccurate estimates of the numerators and denominators of eligible participants/non participants for WIC (which is only available up to age 5) and for school meals which are usually available to children of school age (5-18 in many states , 3-18  in a few).  The authors also omit the information that WIC is available to pregnant, lactating, and post partum people. (line 52)

4) The classification of income in this paper is not congruent with the income eligibility standards for SNAP and WIC participation especially in terms of the $50,000 to  $99,000 category.  Eligibility for these programs depends on the Federal Poverty Level which is calculated based not only on income but on household size.  To qualify as being below the Federal Poverty Level a household with income of $50,000 year would have to have 9 members! (https://aspe.hhs.gov/topics/poverty-economic-mobility/poverty-guidelines/prior-hhs-poverty-guidelines-federal-register-references/2021-poverty-guidelines)  Even acknowledging that WIC  and school meal participants can have income up to 185%  of the FPL it is really difficult to understand from the data presented who proportion of this panel is income eligible for  the  programs of interest. Only a household of 5 or more members with in income of  $50,000 would be eligible for WIC or reduced price school meals. (https://fns-prod.azureedge.net/sites/default/files/resource-files/WIC-Policy-Memo-2021-5-IEG.pdf#page=3)

5) In the introduction the authors fail to note that Food Insecurity is also associated with child and maternal health although the Drennan article which they cite  (among many others they don’t cite) makes this association clear.

In summary, perhaps a more valid conclusion from the sample presented here would be that even relatively ethnically/racially and economically privileged families increased food pantry use during the early days of the COVID epidemic – a finding which was highlighted in the media, but has not to my knowledge been presented in peer reviewed journal.

Author Response

Response to Reviewer 2 Comments

  1. The authors seem to be unaware of confounding by indication – it is an oft repeated finding that SNAP participants have higher rates of food insecurity (FI) than non-respondents because it is often FI that drives recipients to jump through the bureaucratic hoops to participate. SNAP benefits were inadequate to purchase even the Thrifty Food Plan in most states for a month’s worth of food before the 15% COVID American Recovery Act increase in Jan 2021 in benefits for those other than poorest who were already receiving maximum allotment for family size, and remains so even after the $.40 per meal per person increase in October 2021, which has been outstripped by inflation in food prices.

Increase in SNAP benefits for millions of families included in the Families First Coronavirus Response Act passed in March 2020. The initial structure of the policy change increasing benefits to the maximum amount meant those with the lowest incomes already at the maximum benefit amount did not see any increase until passage of subsequent relief packages, while pre-existing eligibility requirements meant that some otherwise eligible lawfully present immigrants who arrived in the U.S. within the past five years remained disqualified for the program.    (https://www.cbpp.org/research/food-assistance/usda-states-must-act-swiftly-to-deliver-food-assistance-allowed-by ; https://www.fns.usda.gov/snap/eligibility/citizen/non-citizen-policy)

We agree with the reviewer and have added this statement to the beginning of our limitations (lines 424-426): “The data collected were cross-sectional and therefore we cannot assess temporality of the association between food insecurity and program participation.” We have also added additional details to the introduction section regarding the unintended consequences of the initial structure of the policy change on lines 51-68.

  1. The inclusion of households with parents born outside the US is not described in the description of the Qualtrics panel. Since one in four children in the US have at least one immigrant parent this is a concerning omission.

We agree that it is not ideal that the parent survey did not collect this information. We have included this as a limitation on lines 433.

  1. The authors claim that their panel (which is 73% Non-Hispanic White people) is reflective of the US population as a whole, or of SNAP recipient families particularly in terms of race/ethnicity, is difficult to defend. 2020 Census data show that only 57.8% of US population as a whole identifies as non-Hispanic White. By recent standards (JAMA. doi:10.1001/jama.2021.17579 based on US Preventive Service Task Force standards) “non-White” is not an acceptable classification – racially minoritized is preferred, but this group should be further disaggregated in description of sample (Black, Hispanic or Latinx, Asian Native American.) even if subgroups are too small for analysis. In terms of SNAP participation race/ethnicity also differs considerably from the Qualtric sample. (New USDA Report Provides Picture of Who Participates in SNAP: https://frac.org › blog › new-usda-report-provides-picture)

Thank you for your comment. We have addressed this above in our third and fourth responses to Reviewer 1. Additionally, we have added disaggregated data for the race variable in our Table 1.

  1. By defining children as those less that 18 years without further disaggregation by child age the authors have an inaccurate estimates of the numerators and denominators of eligible participants/non participants for WIC (which is only available up to age 5) and for school meals which are usually available to children of school age (5-18 in many states , 3-18 in a few). The authors also omit the information that WIC is available to pregnant, lactating, and post partum people. (line 52)

We want to clarify that our analysis included all households with children under 18 years of age. Our goal in this paper was to understand how these households, as a group, fared during the first four months of the pandemic compared to the year prior to the pandemic adjusing for key factors like household income, total number of children, race/ethnicity. While we understand that the eligibility criteria for these programs differ by age of the child, as well as household income and size, our analysis was not restricted to eligible households only. To your second point, we have included more information about eligibility for the WIC program on lines 70-76 in our introduction.

  1. The classification of income in this paper is not congruent with the income eligibility standards for SNAP and WIC participation especially in terms of the $50,000 to $99,000 category. Eligibility for these programs depends on the Federal Poverty Level which is calculated based not only on income but on household size. To qualify as being below the Federal Poverty Level a household with income of $50,000 year would have to have 9 members! (https://aspe.hhs.gov/topics/poverty-economic-mobility/poverty-guidelines/prior-hhs-poverty-guidelines-federal-register-references/2021-poverty-guidelines) Even acknowledging that WIC and school meal participants can have income up to 185% of the FPL it is really difficult to understand from the data presented who proportion of this panel is income eligible for the programs of interest. Only a household of 5 or more members with in income of $50,000 would be eligible for WIC or reduced price school meals. (https://fns-prod.azureedge.net/sites/default/files/resource-files/WIC-Policy-Memo-2021-5-IEG.pdf#page=3)

Thank you for your comment. We did not measure household eligibilty in any of the food assistance programs examined. Our goal was to compare the prevalence in self-reported participation during the year prior to the pandemic with participation during the first four months of the pandemic for all households with children. While eligibility can vary on a monthly basis, our data did not allow us to determine that. The parent survey asked for household income categorically in ranges (e.g., <15K, 15-30K, etc.) instead of as a continuous variable. Additionally, after assessing the responses to the “number of people in household” variable, we found that it was not reliable and should not be used in analysis, as respondents inconsistently included themselves in the count of all household members. As a proxy for number of people in household, we used the variable for number of children in household. That said, we believe we have addressed your concern through adjustment for household income and number of children in our models. Even though eligibility could not be determined, we still controlled for income category, number of children in the households, household size, and other household-level variables so that the prevalence estimates are not affected by compositional differences across households.

  1. In the introduction the authors fail to note that Food Insecurity is also associated with child and maternal health although the Drennan article which they cite (among many others they don’t cite) makes this association clear.

In the introduction (lines 41-45), we describe the association between food insecurity and physical and psychosocial health, developmental and academic outcomes, and future health consequences of children. We have included additional references in our introduction on line 42.

  1. In summary, perhaps a more valid conclusion from the sample presented here would be that even relatively ethnically/racially and economically privileged families increased food pantry use during the early days of the COVID epidemic – a finding which was highlighted in the media, but has not to my knowledge been presented in peer reviewed journal.

Thank you for this thoughtful comment. We expanded the discussion to include this consideration.

Round 2

Reviewer 1 Report

The authors' adequately addressed all prior comments noted.

Author Response

Thank you again to Reviewer 1 for taking the time to look over our paper!

Reviewer 2 Report

Unfortunately in  spite of  an apparently good faith effort to address previous reviewer concerns in the limitations section this article remains seriously flawed and as the underbrush has been cleared from some portions other issues become apparent.  The acknowledged and unacknowledged limitations related to the data collection methods, analytic approach, and the size, and characteristics of the sample and other issues are sufficiently serious as to suggest that this article is not a useful advance in the field of study of the COVID epidemic and food security among families with children.

 There remain some simple fixes, but these fixes   may not be sufficient to warrant publication in the context of the overall difficulties. Nevertheless these should be pointed out.  It is not specified whether p values are one or two tailed.  The findings are not reported in the context of supporting or not supporting the initial hypotheses.  Throughout “race” should be referred to as race/ethnicity and the term “non-White” is not adherent to recent guidelines which were provided in the previous review.

Lines 263-271 do not seem to refer to any analyses elsewhere in the paper that demonstrate statistical testing of whether there are statistical differences in degree  of change in recalled food insecurity between SNAP participants and non-participants, even though as the authors points out this finding if valid is important

Sor

Author Response

Thank you again to Reviewer 2 for taking the time to look over our paper. Please see our responses below.

It is not specified whether p values are one or two tailed. 

We have indicated that our analyses use 2-tailed p-values.

The findings are not reported in the context of supporting or not supporting the initial hypotheses.  

We have added more detail to our discussion to address the initial hypotheses. Please see lines 261 and 268.

Throughout “race” should be referred to as race/ethnicity and the term “non-White” is not adherent to recent guidelines which were provided in the previous review.

We have clarified the word “race” as referring to race/ethnicity on line 114. We have used race instead of race/ethnicity throughout the paper for brevity.

Lines 263-271 do not seem to refer to any analyses elsewhere in the paper that demonstrate statistical testing of whether there are statistical differences in degree  of change in recalled food insecurity between SNAP participants and non-participants, even though as the authors points out this finding if valid is important

Thank you for your comment. Lines 263-267 refer to Figure 1a, which is supported by statistical analyses. We have added additional details on lines 268-283 to clarify our results.